Prognostic significance of SH2D5 expression in lung adenocarcinoma and its relation to immune cell infiltration

Zhou Hao
Li Shengjun
Lin Yuansheng linys202012@163.com
Department of Emergency and Critical Care Medicine, Suzhou Hospital, Affiliated Hospital of Medical School, Nanjing University , Suzhou , Jiangsu , China
Zhang Xin
Electronic publication date: 2023 May 9
Publication date: 2023
Volume: 11
Electronic Location ID: e15238
Received 2022 Sep 21; Accepted 2023 Mar 28
Copyright: ©2023 Zhou et al.
Copyright year: 2023
Copyright holder: Zhou et al.
License: This is an open access article distributed under the terms of the Creative Commons Attribution License, which permits unrestricted use, distribution, reproduction and adaptation in any medium and for any purpose provided that it is properly attributed. For attribution, the original author(s), title, publication source (PeerJ) and either DOI or URL of the article must be cited.
License URL: https://creativecommons.org/licenses/by/4.0/

Keywords: SH2D5, Prognosis, Immune infiltration, Lung adenocarcinoma

Funding: The authors received no funding for this work.

==============================
Objective

Through analyzing the SH2D5 expression profiles, clinical features, and immune infiltration in lung adenocarcinoma (LUAD), the study was intended to discuss the correlations of SH2D5 with prognosis and immune infiltration in LUAD.

Methods

We downloaded transcriptome and clinical data of LUAD patients from TCGA, GEO, and CCLE databases. Sangerbox, R language, GEPIA, UALCAN, and Kaplan-Meier Plotter were adopted to analyze the SH2D5 expression patterns, prognosis, and clinical features. Spearman correlation analysis was performed to determine the association between SH2D5 expression and immune cell infiltration and immune checkpoint genes. The miRNA-SH2D5 relations were predicted by miRDB and starbase. Lastly, quantitative PCR, IHC and Western blot were implemented for validation.

Results

A prominent up-regulation of SH2D5 was noted in the LUAD group relative to the normal group, which was validated by quantitative PCR, IHC and Western blot. SH2D5 expression was inversely related to overall survival (OS) of LUAD patients as well as B cell immune infiltration. Additionally, SH2D5 expression was negatively correlated with dendritic cells resting (p < 0.001), plasma cells (p < 0.001), mast cells resting (p = 0.031) and T cells CD4 memory resting (p = 0.036) in LUAD patients with abundant SH2D5 expression correlated with poor prognosis. Furthermore, enrichment analysis suggested that SH2D5 was associated with lung cancer and immunity. Lastly, we investigated the relationship between the expression of SH2D5 and the use of antitumor drugs.

Conclusion

High SH2D5 expression shares an association with unfavorable prognosis in LUAD, and SH2D5 may also provide new ideas for immunotherapy as a potential therapeutic target.

Introduction

Lung cancer represents a major cause of cancer-specific mortality and thus gives rise to a global health problem (Matsuoka et al., 2018). Lung adenocarcinoma (LUAD), the most frequently occurred histologic pattern of lung cancer, can be sub-classified into acinar, squamous, solid, papillary, and micropapillary tumors, and its prognosis is generally unfavorable owing to early metastasis with an average 5-year survival rate under 20% (Lin et al., 2016; Xu et al., 2020). Despite many therapeutic methods available for LUAD, such as chemotherapy, radiotherapy, targeted therapy, etc., the tumor progresses rapidly and has a high mortality rate (Jones et al., 2021; Siegel et al., 2022). Therefore, it is of great importance to seek applicable immune-associated prognostic markers for LUAD.

SH2 domain-containing protein 5 (SH2D5) is localized on human chromosome 1 and expressed intracellularly and may be involved in regulating synaptic plasticity via controlling the Rac-GTP level (Gregory et al., 2006). As a mammalian-specific adaptor-like protein, SH2D5 possesses an N-terminal PTB domain and a C-terminal SH2-like domain and is abundantly present in the brain at the transcriptional level. SH2D5 can bind to the breakpoint cluster region (BCR) protein and mediate the Rac1 -GTP level (Gray et al., 2014; Liu & Nash, 2012; Um et al., 2014). A meta-analysis has pointed out the relevance of DNA methylation CpG sites to SH2D5 and NPTX2, both of which engage in the modulation of synaptic plasticity, and the functional enrichment analysis has identified the processes such as calcium-binding and estrogen receptor pathways (Winsvold et al., 2018). It has been unraveled that SH2D5 expresses at a higher level in liver tissues sourced from patients with HBV-related hepatocellular carcinoma (HCC) relative to adjacent non-tumor tissues. In addition, HBV infection elevates the SH2D5 levels, and HBx crucially functions in inducing SH2D5 expression; moreover, HBx can stimulate the SH2D5 expression via the NF-B and c-Jun kinase pathways (Zheng et al., 2019). Few studies are focusing on SH2D5 currently. Although SH2D5 is linked to the mechanism of HCC progression, the expression pattern and mechanism of SH2D5 in LUAD remain to be clarified.

Here, we initially determined the SH2D5 expression in LUAD utilizing Sangerbox and R language and subsequently assessed the prognostic value of SH2D5 and its relation to clinical characteristics of LUAD patients by means of Kaplan–Meier, gene expression profiling interactive analysis (GEPIA), and UALCAN analyses. Meanwhile, tumor immune estimation resource (TIMER) was adopted for correlation analysis between SH2D5 expression and immune infiltration in LUAD. Our study highlighted the link between high SH2D5 expression and poor prognosis of LUAD. Furthermore, this study proposed a mechanism that SH2D5 overexpression affects the prognosis of LUAD through controlling immune infiltration.

Materials and Methods

Data collection

All mRNA transcriptome data and clinical data from TCGA, GEO and CCLE database. We attained the clinical data of 594 cases, consisting of 535 LUAD and 59 normal tissue samples. A total of three, 6-week-old mice were purchased from the Shanghai Jihui experimental animal breeding Co., Ltd. (Shanghai, China) (n = 3 in LUAD and paracancerous group). This study was approved by the Laboratory Animal Ethics Committee, Suzhou Institute of Biomedical Engineering and Technology, Chinese Academy of Sciences. Ethical Approval Number: 2022-B27.

UALCAN analysis

The clinical features of SH2D5 in LUAD were analyzed with the assistance of the UALCAN website (http://ualcan.path.uab.edu/index.html) (Chandrashekar et al., 2017). LUAD type-, gender-, lymph node metastasis (LNM)-, smoking-, and stage-based patient stratification was conducted.

GEPIA

The GEPIA database (http://gepia.cancer-pku.cn/) is a newly developed web server based on the TCGA and GTEx projects (Tang et al., 2017), comprising the RNA sequencing expression data of 9736 tumor samples and 8587 normal samples. This database was adopted for survival analysis concerning SH2D5 in this study.

Survival analysis

The prognosis of LUAD based on SH2D5 expression was analyzed by the Kaplan–Meier (KM) website (http://kmplot.com/) (Györffy et al., 2010). A significant difference was noted if p < 0.05.

Immune infiltration analysis

The infiltration status of 22 immune cell types in the LUAD tissues was assessed by ssGSEA. The CIBERSORT algorithm can detect the composition of invasive immune cells in each specimen. The CIBERSORT algorithm allows a machine learning approach to distinguish 22 human immune cell phenotypes with a high degree of specificity and sensitivity (Chen & Wang, 2020). We used the CIBERSORT algorithm to compare the relative percentage of immune cells between the high expression and low expression group. Then, Spearman’s correlation analysis between SH2D5 expression and immune checkpoints.

SH2D5 and drug response

The CellMiner database (https://discover.nci.nih.gov/cellminer/home.do) is developed to help integrate and study molecular and pharmacological data for 60 diverse cancer cell lines in human. SH2D5 expression data was extracted from the CellMiner, and the correlation coefficient between SH2D5 and drugs was calculated with the help of the R core function and visualized by R ggplot2.

Establishment of LUAD in nude mice

Three 6-week-old mice were procured from Shanghai Jihui Laboratory Animal Care Co., Ltd. (Shanghai, China), with LUAD (n = 3) and adjacent non-tumor tissues (n = 3) collected. All mice were kept in an animal facility under pathogen-free conditions. Animal room was maintained at 18–23 °C, 10 h–14 h light and dark cycle. All food, water and other items that meet the rats are sterile. ADCre recombinant adenovirus was supplied by microbix. After pentobarbital sodium anesthesia of 6-week-old mice (45 mg/kg ip), approximately 125 µL of AdCre:CaPi co-precipitates was dripped into the mouse nasal cavity (twice a day, every 5 days, 42 days in total). Mice were euthanized 42 days following induction. Animal euthanasia complies with the principles stipulated by China’s National Standards for Ethical Review of Animal Welfare. Recommended procedures for CO2 euthanasia: the initial flow of CO2 was maintained at 20% to 30% V/min until unconsciousness, and then the flow was intensified (Bremnes et al., 2016). Besides, the CO2 flow should be maintained for at least 1 min after clinical death to avoid reversal. Lastly, cervical dislocation was performed after CO2 euthanasia.

Real-time quantitative PCR (qRT-PCR)

The total RNA extracted from mouse tissues by Trizol (Invitrogen, Waltham, MA) was reverse-transcribed into cDNA with the utility of M-MLV reverse transcriptase (Promega, Madison, WI, USA) and miRNA 1st Strand cDNA Synthesis Kit (by stem-loop) (Vazyme, Nanjing, China). Amplification was conducted employing SYBR Premix Ex Taq kit (TaKaRa, Shiga, Japan) and Applied Biosystems (ABI) StepOnePlus real-time PCR system (ABI, USA). Actin served as a loading control, and qRT-PCR results were analyzed by the 2−ΔΔCt method. The primers for experimental use were as follows: Rat forward: 5-CAAGTCTGAGGCGGAATT-3, reverse: 5-GCAGTGGAGTGATGGTAG-3.

Human SH2D5: forward: 5-TGTGGTTTGGTGGCTGCCTTG-3, reverse: 5-CGACGGTTCTGCGTGGACTGAC-3. miRNA-339-5p: forward: 5-AACACGTGTCCCTGTCCTCCA-3, reverse: 5-ATCCAGTGCAGGGTCCGAGG-3, RT:5-GTCGTATCCAGTGCAGGGTCCGAGGTATTCGCACTGGATACGACCGTGAG-3. All qRT-PCR procedures were conducted in duplicate or more.

Western blotting

Total proteins from cell lines were extracted using a lysis solution (RIPA) bufer, phenylmethylsulfonyl fuoride (PMSF) in a ratio of 100:1. After the protein concentration was determined, the protein was transferred to polyvinylidene fluoride (PVDF) membrane (0.45 µm; Millipore, USA), activated by treatment with methanol for 5 min. A primary antibody (SH2D5, Immunoway, USA) was applied at 4 °C overnight, followed by a secondary antibody.

Immunohistochemical (IHC) assay

We selected three human LUAD tissue samples from Suzhou Science & Technology Town Hospital. This study was approved by the Ethics Committee of Suzhou Science&Technology City Hospital (Ethics Committee No.: IRB2021054). The patient agrees and signs the informed consent form. Primary antibodies used were as follows: A primary antibody (SH2D5, Immunoway, USA). The immunohistochemical results of SH2D5 were evaluated by intensity score combined with staining intensity and area. The intensity score range is 0–9, 0 is no expression, and 9 is high expression.

Statistical analysis

Data were processed employing GraphPad Prism 8 software and statistical analysis was implemented with R software v3.1.3. Measurement data were analyzed by t-test. Cox regression multivariate analysis was carried out for survival analysis. A value of p < 0.05 was indicative of a statistically significant difference.

Results

Abundant SH2D5 expression in LUAD

The study design encompassed SH2D5 expression, SH2D5 characterization, and mechanistic analyses, with the schedules summarized in Fig. 1A. Sangerbox website and R language were applied for SH2D5 expression profiling in different cancers. Relative to non-cancerous tissues, SH2D5 expression was elevated in most tumors, such as bladder urothelial carcinoma, cholangiocarcinoma, esophageal carcinoma, head and neck squamous cell carcinoma, renal papillary cell carcinoma, LUAD, lung squamous cell carcinoma, etc. (Fig. 1B). R language analysis of the TCGA and GEO data exhibited a noticeable elevation of SH2D5 in the LUAD tissues versus normal tissues (Figs. 1C, 1D and 1E) and additional qRT-PCR assay, Western blot and IHC staining displayed consistent results that SH2D5 expressed at a higher mRNA level in mouse LUAD tissues and BEAS-2B and H1299 cell lines as compared to adjacent non-tumor tissues (Fig. 1F, Figs. S2A–S2C).

Figure 1 Abundant SH2D5 expression in LUAD.

(A) Flow diagram of this studies. (B) Expression level of SH2D5 in pan-carcinomas by sangerbox website. (C–D) The TCGA data show the expression of SH2D5 in the LUAD tissues versus normal tissuesby R language. (E) The GSE43458 dataset from GEO shows the expression of SH2D5 between the LUAD tissues and normal tissues. (F) qRT-PCR assay displayed SH2D5 mRNA level in mouse LUAD tissues and adjacent non-tumor tissues.

Relations of SH2D5 expression to clinical features and prognosis of LUAD

To further characterize the clinical significance of SH2D5, the SH2D5 expression in TCGA-LUAD samples was visualized by UALCAN website analysis, which revealed considerably higher SH2D5 expression in LUAD than in non-cancerous tissues (Fig. 2A, p = 3.03e−08). SH2D5 expression in normal and LUAD groups was suggested to share associations with gender, LNM, smoking, and stage (Figs. 2B–2E). Table 1 shows the comparison of different types of patients in LUAD, and there is no difference in his groups. Next, the relation of SH2D5 expression to the clinical overall survival (OS) of 316 LUAD patients was examined by uni-variable and multi-variable analyses. The results revealed that SH2D5 overexpression was related to worse OS (Table 2). Hence, SH2D5 shows promise as an independent prognostic factor for LUAD.

Figure 2 Relations of SH2D5 expression to clinical features and prognosis of LUAD.

(A) The SH2D5 expression in TCGA-LUAD samples was visualized by UALCANwebsite analysis. (B–E) SH2D5 expression in LUAD was associations with gender, LNM, smoking, and stage.

Table 1 Baseline charateristics of patients with LUAD.

Characteristic	Low expression of SH2D5	High expression of SH2D5	p	
n	267	268		
T stage, n (%)			0.075	
T1	98 (18.4%)	77 (14.5%)		
T2	130 (24.4%)	159 (29.9%)		
T3	28 (5.3%)	21 (3.9%)		
T4	11 (2.1%)	8 (1.5%)		
N stage, n (%)			0.301	
N0	178 (34.3%)	170 (32.8%)		
N1	44 (8.5%)	51 (9.8%)		
N2	32 (6.2%)	42 (8.1%)		
N3	0 (0%)	2 (0.4%)		
M stage, n (%)			0.109	
M0	183 (47.4%)	178 (46.1%)		
M1	8 (2.1%)	17 (4.4%)		
Pathologic stage, n (%)			0.117	
Stage I	155 (29.4%)	139 (26.4%)		
Stage II	62 (11.8%)	61 (11.6%)		
Stage III	37 (7%)	47 (8.9%)		
Stage IV	8 (1.5%)	18 (3.4%)		
Gender, n (%)			0.463	
Female	138 (25.8%)	148 (27.7%)		
Male	129 (24.1%)	120 (22.4%)		
Age, n (%)			0.377	
< =65	121 (23.4%)	134 (26%)		
>65	135 (26.2%)	126 (24.4%)		
Smoker, n (%)			0.762	
No	39 (7.5%)	36 (6.9%)		
Yes	220 (42.2%)	226 (43.4%)		

Table 2 Univariate analysis and Multivariate analysis of the correlation of SH2D5 expression with OS among lung adenocarcinoma.

Parameter	Univariate analysis	Multivariate analysis	
	HR	HR.95L	HR.95H	p value	HR	HR.95L	HR.95H	p value	
Age	1.00	0.98	1.02	0.84					
Gender	1.04	0.72	1.49	0.85					
Stage	1.65	1.40	1.95	2.58E−09	1.37	1.09	1.73	7.29E−03	
T	1.63	1.32	2.02	8.60E−06	1.24	0.89	1.56	0.07	
M	1.76	0.96	3.20	0.07					
N	1.79	1.46	2.20	2.41E−08	1.34	1.02	1.74	3.37E−02	
SH2D5	1.36	1.08	1.71	9.48E−03	2.16	1.46	3.18	1.03E−04	
Notes.

Bold values indicate p < 0.05.

HR hazard ratio

CI confidence interval

The prognostic significance of SH2D5 in LUAD

Given the differential expression of SH2D5 in LUAD, the study focus was shifted onto the relevance of SH2D5 expression to prognosis. LASSO regression analysis on SH2D5 expression was implemented with the utility of the TCGA dataset, and the patients were sub-classified into high and low expression groups. The gene expression, survival time, and survival status are depicted in Fig. 3A. KM survival analysis offered data showing the association between high SH2D5 expression and poor prognosis (Fig. 3B). After validation through the KM-plot, GEPIA, and UALCAN databases, consistent results were generated (Figs. S1A–S1D). Then, receive operating characteristic curve (ROC) involving SH2D5 were plotted and area under curve (AUC) values were estimated, with the results indicating certain diagnostic efficacy of SH2D5 for LUAD (Fig. 3C, Fig. S1E).

Figure 3 The prognostic significance of SH2D5 in LUAD.

(A) LASSO regression analysis on SH2D5 expression was implemented with the utility of the TCGA dataset. (B) KM survival analysis showed the association between SH2D5 expression. (C) ROC curves showed diagnostic efficacy of SH2D5 for LUAD.

Prognostic value of SH2D5 expression correlates with immune infiltration in LUAD

The infiltration status of 22 immune cell types in the LUAD tissues was assessed by ssGSEA. The relationship between SH2D5 expression and immune cell infiltration was evaluated by Spearman correlation analysis. SH2D5 expression was negatively correlated with dendritic cells resting (p < 0.001), plasma cells (p < 1.001), Mast cells resting (p = 0.031) and T cells CD4 memory resting (p = 0.036), and positively correlated with macrophages M0 (p < 0.001), NK cells resting (p < 0.001), T cells CD4 memory activated (p < 0.001) and macrophages M1 (p = 0.001) (Fig. 4A). We then investigated the expression of SH2D5 in 22 immune cell types between high and low expression groups. The results showed plasma cells, T cells CD4 memory resting, T cells CD4 memory activated, NK cells resting, macrophages M0 were significant between high and low expression groups (Fig. 4B). Furthermore, Among the immune checkpoints, CD40LG and TNFSF15 presented a negative correlation with the immune infiltrates, while CD276 and CD70 presented a positive correlation (Fig. 4C). We demonstrate that SH2D5 expression levels correlate with immune cell infiltration in LUAD of the TCGA dataset (Fig. 4D).

Figure 4 SH2D5 relates to immune infiltration level in LUAD.

(A) Spearman’s correlation analysis between infiltration levels of 22 immune cell types and SH2D5 expression levels in the LUAD tissues. (B) The ratio of 22 immune cells types in LUAD tissues in the SH2D5 high and low expression groups. (C) Correlation between SH2D5 expression and immune checkpoints. (D) The correlation analysis between the expression levels of SH2D5 and theexpression levels of dendritic cells resting, plasma cells, T cells CD4 memory resting, macrophages M0 and NK cells resting.

Immune infiltration has shown a relation to the prognosis of lung cancer (Conway et al., 2016; Fan et al., 2021; Sun et al., 2020b). We have demonstrated the relation of elevated SH2D5 expression to poor prognosis and also found the association between SH2D5 and immune infiltration. Thus, we surmised that the prognostic performance of SH2D5 in LUAD was potentially relevant to immune infiltration. We subsequently performed a prognostic analysis employing the KM website to correlate SH2D5 expression with immune cell subsets in LUAD. The data illustrated a poor prognosis of LUAD with high SH2D5 expression and enrichment of B cells, CD4+ T cells, macrophages, and Treg cells (Figs. 5A–5H). The aforementioned data, therefore, suggest that SH2D5 abundance in LUAD may affects prognosis through immune infiltration.

Figure 5 Prognostic value of SH2D5 expression correlates with immune infiltration in LUAD.

(A–H) The prognostic performance of SH2D5 in LUAD was potentially relevant to immune infiltration.

Network enrichment analysis of SH2D5 in LUAD

For the purpose of assessing the biological significance of SH2D5 in LUAD, the co-expression patterns of SH2D5 in LUAD were examined in the CCLE dataset (Fig. 6A). Gene Ontology (GO) analysis of 12 co-expression genes exhibited their correlations with natural killer (NK) cells, mediation of cytotoxicity, activation of molecular adapters, etc. Through Kyoto Encyclopedia of Genes and Genomes (KEGG) pathway analysis, the aforementioned genes were relevant to the positive regulation of natural killer cells, phagocytosis-mediated immune regulation, and regulation of phagocytosis (Fig. 6B). Then, we used GSEA between high and low SH2D5 expression datasets. 6 hallmark gene-sets (including KEGG_COLORECTAL_CANCER, KEGG_MAPK_SIGNALING_PATHWAY, KEGG_NON_SMALL_CELL_LUNG_CANCER, KEGG_PATHWAYS_IN_CANCER, KEGG_PRIMARY_IMMUNODEFICIENCY, KEGG_SMALL_CELL_LUNG_CANCER) were chosen for analysis (Fig. 6C). The results showed that non-small cell lung cancer, pathways in cancer, primary immunodeficiency are differentially enriched in SH2D5 high expression phenotype of LUAD.

Figure 6 Network enrichment analysis of SH2D5 in LUAD.

(A) The co-expression patterns of SH2D5 in LUAD were examined in the CCLE dataset. (B) GO and KEGG terms of SH2D5-related genes. (C) Gene set enrichment analysis (GSEA) of the altered signaling pathways in the LUAD tissues based on the SH2D5-associated DEGs between the high- and low-SH2D5 expression groups. (D) miRDB and starbase tools were used to predict the target of SH2D5. (E) miRDB and RNA22 were employed for prediction of miRNA-339-5p binding site within BTN3A2 3′ UTR.

To seek SH2D5 upstream molecules in LUAD, we predicted the microRNAs (miRNAs) binding to SH2D5 through miRDB and starbase tools and found SH2D5 to be the target of miRNA-339-5p (Fig. 6D). We further verified the expression levels of miRNA-339-5p in BEAS-2B and H1299 (Fig. S2D). Then, the databases miRDB (http://www.mirdb.org/) and RNA22 (https://cm.jefferson.edu/rna22v2.0/) (Miranda et al., 2006) were employed for prediction of miRNA-339-5p binding site within SH2D5 3′UTR (Fig. 6E). This provides a potential mechanism that miRNA-339-5p may target SH2D5.

SH2D5 and drug response

SH2D5 expression exhibited positive associations with the drug reactions of Staurosporine, while negative associations with anti-cancer drugs Crizotinib, Tamoxifen, Dolastatin 10, Vinorelbine, Vinblastine, Eribulin mesilate, Homoharringtonine, Tyrothricin (Fig. 7).

Figure 7 Showing the relationship between SH2D5 expression and expected drug response.

Discussion

Up-regulation of SH2D5 in LUAD was determined in this study by Sangerbox and R language. Our study additionally unveiled the relevance of SH2D5 expression to LUAD prognosis. Furthermore, the study suggests a mechanism wherein elevated SH2D5 expression may affect the prognosis of LUAD through immune infiltration.

SH2D5 has been documented to be up-regulated in liver tissues of HBV-HCC patients as compared to adjacent non-tumor tissues, and gain- and loss-of-function experiments have substantiated that elevation of SH2D5 expedites liver cancer cell proliferation in vitro and in vivo, and patients with high SH2D5 RNA levels have shorter OS (Zheng et al., 2019). Consistent with this finding, the present study also suggested the notably elevated SH2D5 expression in the LUAD samples compared with normal samples (Figs. 1C and 1D). Our KM survival analysis also linked the high SH2D5 expression to unsatisfied prognosis (Fig. 3B), which was validated by uni-variable and multi-variable analyses (Table 2). Hence, SH2D5 may serve as an independent prognostic indicator for LUAD and may become an appealing therapeutic target for inhibiting LUAD cell proliferation.

A recent study has expounded that elevation of doublecortin-like kinase 1 (DCLK1) predicts clinical prognosis in lung and gastric cancer patients, showing an association with immune infiltration (Wu et al., 2020; Yan et al., 2022). High expression of C-type lectin domain family 3 member B (CLEC3B) correlates with favorable OS, which may be linked to immune infiltration and immune activation in lung cancer (Lu et al., 2022; Sun et al., 2020a). GBE1 knockdown can promote the secretion of CCL5 and CXCL10, and the recruitment of CD8+ T lymphocytes into the tumor microenvironment, suggesting that GBE1 may be an important target to inhibit the progression of LUAD tumor through immunotherapy (Li et al., 2019). THBS2+ CAFs interact with macrophages and CD8+ T and B lymphocytes in the early LUAD tumor microenvironment, and high THBS2 in LUAD is associated with reduced immune cell infiltration and elevated immune exhaustion markers (Yang et al., 2022). In LUAD, the high expression of chemokine CCL17 promotes local immune infiltration and anti-tumor immune response, which may contribute to better survival and prognosis of early LUAD patients (Ye et al., 2022). Many of the studies mentioned above have linked immune infiltration to lung cancer survival and prognosis. Our analyses on SH2D5 expression, prognostic value, and LUAD immune infiltration showed that SH2D5 was strongly inversely correlated with Immune infiltration in LUAD and that SH2D5 correlated with unfavorable prognosis of LUAD where B cells, CD4+ T cells, macrophages, and Treg cells were enriched. These results collectively suggest the potential of SH2D5 abundance in LUAD to affect prognosis via modulating immune infiltration.

miRNA-339-5p has recently been substantiated to exert a vital role in a diversity of cancers (Hu et al., 2018; Luo et al., 2019), yet the molecular mechanism and significance of miRNA-339-5p in the progression of LUAD are still undefined. Prior studies have illustrated that miRNA-339-5p was downregulated and miRNA-339-5p gain-of-function considerably impedes the invasiveness and migration of LUAD cells (Li et al., 2018a; Li et al., 2018b). It has been further proposed that miRNA-339-5p is poorly expressed in LUAD, and SH2D5 acts as a target of miRNA-339-5p, which is coincident with our findings. We consider that miRNA-339-5p/SH2D5 may be another mechanism of LUAD development. Additional experiments and validation of this hypothesis are required.

In the meantime, some limitations exist that we fail to deeply explore the relevance of SH2D5 to tumor invasion due to the database and that in vivo and in vitro experiments are warranted for additional validation of our findings.

Conclusion

High SH2D5 expression shares an association with unfavorable prognosis in LUAD, and SH2D5 may also provide new ideas for immunotherapy as a potential therapeutic target.

Supplemental Information

Supplemental Information 1 PCR data

Click here for additional data file.

Data S1 Raw data

Click here for additional data file.

Figure S1 SH2D5 is associated with prognosis

A.D. The correlation between SH2D5 and prognosis was verified by KM-plot, GEPIA, and UALCAN databases. F. ROC curves verified diagnostic efficacy of SH2D5 for LUAD by R.

Click here for additional data file.

Figure S2 The expression level of SH2D5 and miRNA-339-5p

(A) Relative SH2D5 expression levels in BEAS-2B and H1299 cell lines. (B) Western blot analysis of the protein expression of SH2D5 overexpression in BEAS-2B and H1299 cell lines. The left side shows the Western blotting result and the right side shows the statistical diagram. (C) Relative miRNA-339-5p expression levels in BEAS-2B and H1299 cell lines.

Click here for additional data file.

File S1 Code and dataset

Click here for additional data file.

Supplemental Information 6 Author Checklist

Click here for additional data file.

Supplemental Information 7 Full-length uncropped blots

Click here for additional data file.

Abbreviations

LUAD Lung adenocarcinoma

SH2D5, SH2 domain-containing protein 5

OS overall survival

CNV copy number variance

ROC receiver operating characteristic curve

TCGA The Cancer Genome Atlas

AUC area under the curve

Additional Information and Declarations

Competing Interests

Author Contributions

Human Ethics

Animal Ethics

Data Availability

The authors declare there are no competing interests.

Hao Zhou performed the experiments, prepared figures and/or tables, and approved the final draft.

Shengjun Li analyzed the data, authored or reviewed drafts of the article, and approved the final draft.

Yuansheng Lin conceived and designed the experiments, prepared figures and/or tables, and approved the final draft.

The following information was supplied relating to ethical approvals (i.e., approving body and any reference numbers):

The Medical Ethics Committee of Suzhou Science and Technology City Hospital approved the research (IRB2021054).

The following information was supplied relating to ethical approvals (i.e., approving body and any reference numbers):

The Laboratory Animal Ethics Committee of Suzhou Institute of Biology and Engineering Technology, Chinese Academy of Sciences approved the study (2022-B27).

The following information was supplied regarding data availability:

The raw data is available in the Supplemental Files.

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
