# Peer review of "Prognostic significance of SH2D5 expression in lung adenocarcinoma and its relation to immune cell infiltration"

_PeerJ, doi:10.7717/peerj.15238_

## Round 0.1 · original submission · Major Revisions

All three reviewers gave their opinions, and I basically agree with them. Authors must carefully revise and add relevant experiments or analysis, otherwise the reviewer has the right to reject the manuscript.

Reviewer 1 ·

Basic reporting

There are some questions in writing. The language of the article needs a thorough and careful revision. For examples, line 180, "sh2d5" and "luad".

Experimental design

1. To confirm the result of elevated SH2D5 expression of LUAD in TCGA and GEO, the author established a mouse lung cancer model, and detected the SH2D5 mRNA expression in cancer tissues and adjacent non-tumor tissues. However, it is not the most appropriate method for this situation. I suggest that the authors collect several human lung adenocarcinoma tissues and corresponding adjacent tissues, and detect the changes of SH2D5 mRNA or protein expression level by qPCR, western blot or/and IHC staining.

2. Line 96-98. The description of the “immune infiltration analysis” method needs more detail.

3. Line 192-193. "For the purpose of assessing the biological significance of SH2D5 in LUAD, the co-expression patterns of SH2D5 in LUAD were examined in the CCLE dataset." Why did the authors use CCLE dataset (a cell line dataset) instead of the TCGA LUAD dataset (a clinical sample dataset)? In addition, did the author use the data of the entire CCLE dataset? Or the data of lung adenocarcinoma cell lines in CCLE dataset were selected for analysis?

4. Line 148-152 and Figure 2. The author mentioned that “High SH2D5 expression in LUAD was suggested to share associations with gender, 152 LNM, smoking, and stage.” However, the authors only compared the SH2D5 expression of normal tissues and LUAD tissues. I suggest that the authors should compare SH2D5 expression among lung cancer samples of different genders, N stages, smoking status or stages.

5. Line 206-211 and Figure 6C&D. The authors found that SH2D5 may be the target of miRNA-339-5p. The authors need to conduct several experiments to confirm this prediction in LUAD cell lines.

Validity of the findings

no comment

Additional comments

no comment

·

Basic reporting

1.The figures quality was so poor that I could barely make out the text in it. Please improve.
2.There are some word case errors(for example 11 pages with 180 lines). There are some grammatical errors in the text, which need to be corrected. The manuscript requires English polishing.

Experimental design

The experimental design is consistent with the purpose and scope of the journal

Validity of the findings

The topic of this manuscript is interesting

Additional comments

Thanks for giving me the opportunity to review this manuscript. This study aimed to analysis the effect on LUAD by the expression of SH2D5. The topic of this manuscript is interesting. I have some specific comments below.

1. The figures quality was so poor that I could barely make out the text in it. Please improve。
2.Data sources, including animal experiments. But in 1.1, it is not mentioned. Please add
3.In the manuscript, the author proposes:High SH2D5 expression in LUAD was suggested to share associations with gender, LNM, smoking, and stage (Table 1, Figure 2B-2E). I don't agree with this statement. From Table 1, the expression of SH2D5 does not differ from the grouping of these characteristics. Please correct
4.There are some word case errors(for example 11 pages with 180 lines). There are some grammatical errors in the text, which need to be corrected. The manuscript requires English polishing.
5.An additional paragraph should be added to the discussion. The authors analyzed how immune cells affect SH2D5 expression through immune function, thereby affecting the prognosis of LUAD.

Reviewer 3 ·

Basic reporting

Using open access data, such as TCGA or GEO databases, to analyze and screen tumor differential genes is a popular approach. Combined with the results of the LUAD mice model, the authors found that SH2D5 expression was closely related to the prognosis of LUAD. However, there are still some details and experiments necessary to refine the findings.

Experimental design

no comment

Validity of the findings

1. The results in LUAD mice are remarkable, so the model must be validated by HE staining or IHC. (ps: Tissue section staining was described in the method, but it was not shown in the results of the article. Please explain the reason.)
2. Although the authors showed a robust increase in mRNA levels of SH2D5, the results of changes in protein levels were more convincing in the LUAD mice. So, IHC staining or WB validation is recommended. (ps: An alternative approach is to perform IHC on tissue sections from patients)
3. The expression of miR-339-5p should be validated in the LUAD mice model.

Additional comments

4. The diagram in Figure 6D is puzzling. Because the miRNA is paired complementally with the 3 'UTR of the target gene, which is usually an RNA-to-RNA interaction.
5. In the discussion section, the authors should propose a reasonable hypothesis for the mechanism of SH2D5 that regulates immune infiltration and immune activation, which is database-independent.
6. Please note the formatting of the gene and make sure these references are correct. For example, lines 180 and 225...

---

## Round 0.2 · Major Revisions

I think the reviewer's requirements are reasonable. Please be sure to answer relevant questions directly and supplement relevant experimental data.

·

Basic reporting

Clear language expression.

Experimental design

Reasonable experimental design.

Validity of the findings

This study has certain implications for clinical and future.

Additional comments

No

Reviewer 3 ·

Basic reporting

In the revised version, the authors add some experiments and discussions to illustrate the role of SH2D5 in lung adenocarcinoma. However, some experiments or details still need to be improved.

Experimental design

no comment

Validity of the findings

no comment

Additional comments

1. Although the authors added experiments that attempted to illustrate the high expression of SH2D5 in lung adenocarcinoma cell lines, there were individual differences in direct comparisons between cell lines. I still think IHC staining from patients or model mice is necessary.
2. In Figure 6D, the RNA sequences usually contain U instead of T.

---

## Round 0.3 · Minor Revisions

The multivariate analysis method in Table 2 is not optimal. Please refer to relevant tutorials for re-analysis. Generally, only factors that are meaningful in univariate analysis are selected for multivariate analysis.

---

## Round 0.4 · accepted · Accept

There were no obvious defects in the research content, which basically met the publication requirements.